# Pediatric Myxedema Due to Autoimmune Hypothyroidism: A Rare Complication of a Common Disorder

**DOI:** 10.3390/children10040614

**Published:** 2023-03-24

**Authors:** Elisa Bonino, Patrizia Matarazzo, Raffaele Buganza, Gerdi Tuli, Jessica Munarin, Claudia Bondone, Luisa de Sanctis

**Affiliations:** 1Department of Pediatric Endocrinology, Regina Margherita Children’s Hospital, Città della Salute e della Scienza, 10126 Torino, Italy; 2Department of Public Health and Pediatric Sciences, University of Turin, 10126 Torino, Italy; 3Postgraduate School of Pediatrics, University of Turin, 10126 Torino, Italy; 4Department of Pediatric Emergency, Regina Margherita Children’s Hospital, Città della Salute e della Scienza, 10126 Torino, Italy

**Keywords:** myxedema, hypothyroidism, growth delay, pituitary hyperplasia

## Abstract

In children, hypothyroidism usually presents non-specific symptoms; symptoms can emerge gradually, compromising a timely diagnosis. We report the case of a 13-year-old male, who was admitted to the hospital due to swelling of the torso and neck. Besides these symptoms, the child was healthy, except for a significant growth delay. Ultrasound evaluation and blood tests led to the diagnosis of myxedema secondary to severe hypothyroidism, which was due to autoimmune thyroiditis. Further investigations revealed pericardial effusion and pituitary hyperplasia, with hyper-prolactinemia. Treatment with levothyroxine led to edema regression and clinical, hemato-chemical and radiological improvement. After 6 months, growth velocity increased, although the recovery of growth already lost was not guaranteed. Brain MRI showed regression of pituitary hyperplasia. The diagnostic delay in this case was probably due to the patient’s apparent good health, and the underestimation of growth restriction. This report underlines the importance of growth monitoring in adolescence, a critical period for identifying endocrine conditions; if undiagnosed, these conditions can lead to serious complications, such as myxedema in hypothyroidism, with potential effects beyond growth on multiple organs.

## 1. Introduction

Hypothyroidism, congenital or acquired, is the most common thyroid disorder in children. It can be caused by primary thyroid disease (primary hypothyroidism), or impaired production of the pituitary/hypothalamic hormones that regulate thyroid function (secondary/tertiary hypothyroidism). Autoimmune thyroiditis is the most common cause of acquired hypothyroidism, and it usually occurs in older children, adolescents or adults. In particular, it is present in 1–2% of adolescents, in early-to-mid-puberty [1]. The clinical presentation varies deeply based on the magnitude of the hormonal deficit and the speed with which it develops. Patients with subclinical hypothyroidism are often asymptomatic. Children with overt disease usually have non-specific clinical signs and symptoms, such as sluggishness, lethargy, cold intolerance, constipation, dry skin, brittle hair, facial swelling and muscle aches. Decreased serum thyroid hormone levels may also have deleterious effects on growth, pubertal development and school performance. In severe untreated hypothyroidism, a rare clinical sign is myxedema, characterized by non-focal thickening and induration of the skin due to increased deposition of connective tissue constituents, such as glycosaminoglycans. It leads to fluid retention and swelling of the subcutaneous tissues [2], causing periorbital, facial, supraclavicular, pretibial edema or generalized swelling. If hypothyroidism is profoundly decompensated, a life-threatening myxedema coma can occur. It is characterized by altered mental status and changes of vital signs, including hypothermia, bradycardia and hypotension, and represents a medical emergency [3].

This paper describes the peculiar case of an adolescent affected by severe hypothyroidism due to autoimmune thyroiditis, diagnosed only after the onset of myxedema, a very rare complication in childhood.

## 2. Case Report

A 13-and-a-half-year-old boy was presented to the emergency department due to extensive swelling, which had appeared 7 days earlier as submandibular swelling, and rapidly worsened despite treatment with ibuprofen and antibiotics.

Family history was unremarkable, except for delayed puberty, reported both in parents and sibling.

The patient’s past history was not contributory until the last few months, when he complained of mild constipation. A more detailed history revealed that the parents noticed slow speech, although not associated with fatigue, as well as difficulty concentrating, impaired school performance and sleep disturbances.

On physical examination, dry, cold skin and significant submandibular, supraclavicular and sub-axillary soft, non-pitting edema were present. The edema were painless and not associated with crepitus on palpation, reasonably ruling out the hypothesis of subcutaneous emphysema. No hydrocele or edema were present in other parts of the body. Heart tones were muffled, heart rate was 60 beats per minute and blood pressure was 102/73 mmHg. Peripheral pulses and capillary refill time were normal, clinically excluding impaired cardiac hemodynamics or peripheral hypoperfusion. The abdomen was swollen, and tender to palpation. The height was not measured in first instance in the emergency room, with priority given to blood chemistry and instrumental investigations.

The investigation of the main renal, gastrointestinal and cardiac causes of edema demonstrated an increase in serum creatinine levels (1.07 mg/dL). A chest X-ray ruled out pleuro-parenchymal lesions and mediastinal enlargement; dimensions of the cardiac shadow were at the upper limits of the reference values. The electrocardiogram was normal, with the voltages in all leads being normal for age. The echocardiography revealeda circumferential pericardial effusion, up to 12 mm, without alterations of cardiac hemodynamics, evaluated by Doppler study. The abdominal ultrasound study (US) detected a small layer of free liquid in the rectovesical pouch.

The US of thyroid and edematous soft tissues showed an enlarged thyroid gland, with heterogeneous echo structure, multiple hypoechoic foci, micronodular appearance and increased vascularization (consistent with thyroiditis), and thickening of subcutaneous adipose tissue, respectively (Figure 1A–C).

The thyroid gland was difficult to palpate due to neck edema, but hormonal investigation revealed extremely low free thyroxine levels (FT4 0.6 pg/mL, normal range 9.8–16.3) with marked increase in thyroid-stimulating hormone levels (TSH > 1000 μIU/mL, normal range 0.51–4.3), and elevated titers of anti-thyroid peroxidase antibodies (361 IU/mL, normal value < 34).

The clinical and laboratory findings suggested a diagnosis of myxedema secondary to severe hypothyroidism in autoimmune thyroiditis, and the boy was admitted to the endocrinology department.

The growth assessment indicated a height of 141 cm (−1.95 SDS according to Tanner standards [4,5]) and weight of 41 kg. At the previous measurement, completed when the patient was 10 and a half years old, height was 138 cm. Comparison between the two measurements showed delayed, almost stunted, growth. The patient’s target height was 178.0 cm (0.25 SDS). During the X-ray, bone age was calculated as 11 years, according to Greulich and Pyle (Figure 2A,B).

The patient’s pubertal development was age-appropriate (testicular volume of 8 mL and Tanner stage 3 for pubic hair).

After excluding associated hypocortisolism (ACTH 14.3 ng/L, cortisol 119.8 mcg/L, normal ranges 3.6–60.5 ng/L and 62–194 mcg/L, respectively), oral replacement therapy with levothyroxine was started. The dose was initially low (25 mcg/day), and then increased gradually over the following days based on close hormonal monitoring. Screening for autoimmune diseases associated with the condition, such as type I diabetes, celiac disease, autoimmune gastritis and autoimmune hypoparathyroidism, was performed and resulted negative.

Considering the extreme elevation of TSH, according to the internal protocol, hypothalamic-pituitary magnetic resonance imaging (MRI) was performed and revealed pituitary hyperplasia (diameters of 16 × 12 × 16 mm, with convex upper margin) (Figure 3).

The pituitary function evaluation disclosed hyperprolactinemia (PRL 118.4 ng/mL, normal range 4–15.2), with no other significant abnormalities on remaining axes (LH 1.6 IU/L, FSH 5.7 IU/L, testosterone 2.32 ng/mL, IGF1 168.2 mcg/L, copeptin 5.5 pmol/L) [normal ranges: LH 1.3–8.6 IU/L, FSH 2–12.5 IU/L, testosterone 0.3–8.05 ng/mL, IGF1 83–519 mcg/L, copeptin 3–8 pmol/L].

An ophthalmological examination ruled out optic chiasm involvement, describing the visual field as normal.

During hospitalization, the patient’s clinical status showed progressive improvement, the edema regressed and the pericardial effusion decreased to 4 mm. At discharge, after 2 weeks, FT4 levels had increased (6.3 pg/mL), while TSH levels (610.9 μIU/mL) and creatinine (0.99 mg/dL) had decreased. Levothyroxine dose at discharge was 75 mcg and 100 mcg every other day. 3 weeks after diagnosis, FT4 was in range (11.1 pg/mL). After 2 months, TSH (1.02 μIU/mL) and creatinine (0.61 mg/dL) also normalized, and prolactin levels decreased significantly (36.2 ng/mL). MRI at 6 months showed resolution of the pituitary hyperplasia, with height reduced from 16 to 3.5 mm. Growth assessment revealed high growth velocity at 6 months (11.6 cm/year) and in the first year of treatment (14.6 cm/year).

Table 1 summarizes the first year of patient follow-up. Currently, the boy is on levothyroxine at 100 mcg/day. Close monitoring of thyroid function, and clinical–biochemical assessment of growth and pubertal development have been planned.

## 3. Discussion

Hypothyroidism in children can develop insidiously, with slowly progressive physical changes and mild symptoms that may remain unrecognized [6]. In our case, despite the patient experiencing a long-lasting condition, only mild constipation and slow speech were reported. No lifestyle changes, sleep disturbance, fatigue or difficulty concentrating were noticed by the parents or the patient himself. Indeed, his performance at school was reasonable before diagnosis. It can be assumed that a very gradual loss of thyroid function allowed for better tolerance to the hypothyroid condition.

The most important sign that led to suspicion of long-standing hypothyroidism was the decreased growth velocity, demonstrated from the age of about 10 and a half years old, which was probably underestimated due to a family history of pubertal delay. In our patient, pubertal development was age-appropriate. Children with hypothyroidism generally have delayed puberty; in rare cases of long-lasting untreated hypothyroidism, precocious puberty secondary to hyperprolactinemia or increased TSH levels [7,8] has also been reported.

It is well-known that any child with impaired linear growth thyroid function should be evaluated, as thyroid hormone deficiency negatively acts on the growth plate, and downregulates growth hormone and insulin-like growth factor 1 (IGF1) secretion [9]. Thus, growth restriction is one of the most common manifestations of pediatric hypothyroidism; however, it tends to be insidious in onset, and may be present for several years before other symptoms occur, if they occur at all. [10]

The diagnostic delay in our case could be partly explained by the good general health of the boy, and the absence of acute conditions that required a pediatric evaluation. Adolescents between 10 and 14 years of age are often lost to medical follow-up, even though it would be essential for an appropriate monitoring of growth and puberty.

In hypothyroid patients, an underlying hypocortisolism has to be ruled out before starting any treatment, since levothyroxine may precipitate adrenal crises in untreated primary adrenal insufficiency [11]. In our patient, normal values of ACTH and cortisol could exclude this condition.

Myxedema is extremely rare, especially in children, and it is reported in only a few works in literature [1,6,12]. Severely advanced hypothyroidism can also lead to myxedema crisis or coma, a life-threatening medical emergency [3,13]; this emergency can be prevented by prompt initiation of levothyroxine treatment.

In our patient, MRI, performed in accordance with the Center’s internal protocol for patients with very high TSH levels, detected a pituitary hyperplasia. The true incidence of this condition remains largely unknown because MRI is not routinely indicated in the diagnostic pathway of primary hypothyroidism, and it may remain radiologically undocumented. The pathogenesis of pituitary hyperplasia in severe long-standing primary hypothyroidism is explained by the reduction of circulating thyroid hormones. It leads to loss of the negative feedback on hypothalamus, with consequent excessive secretion of thyrotropin-releasing hormone (TRH) and proliferation of thyrotropic cells, leading to compensatory hyperplasia [12]. Enlargement of the pituitary gland in these patients can be very pronounced, even with suprasellar extension; however, unlike pituitary tumors or craniopharyngioma, it rarely causes signs or symptoms. Instead, it regresses with levothyroxine treatment and usually does not require surgery [12].

The remaining pituitary function resulted normal, with the only exception of hyperprolactinaemia. The increased prolactin levels, a very common laboratory finding of primary hypothyroidism, are related to the increased production of TRH, which stimulates the secretion of both TSH and prolactin from the pituitary gland, and may cause galactorrhea in children with severe hypothyroidism. Prolactin levels usually normalize after thyroid hormone replacement [12].

Other complications in our patient were transient elevation of serum creatinine and pericardial effusion, not compromising the ventricular function. The increased serum creatinine is frequently reported in hypothyroid patients, and may be due to a transient functional renal impairment, which improves with thyroid replacement [14].

The pericardial effusion, instead, is very uncommon and mostly reported in hypothyroid patients with Down syndrome [15]; it is determined by the increased capillary permeability and disruption of lymphatic drainage. The pericardium’s ability to expand, however, usually prevents cardiac tamponade, and replacement therapy leads to regression of this condition [15,16].

Achievement of euthyroid state is essential to address all the multisystemic effects of severe hypothyroidism. In children, this is especially important as it restores normal growth and pubertal development. Children whose hypothyroidism is promptly diagnosed and treated before puberty typically have good catch-up growth and normal adult height, unless they have other causes of short stature [17]. Conversely, children who have long-standing chronic hypothyroidism often fail to reach their genetic growth potential, and the adult height may be significantly reduced. This can happen, in particular, if the hypothyroidism develops during adolescence, compromising the pubertal growth spurt. In addition, when the treatment is started in this phase of catch-up growth, the bone age advancement and the rapid skeletal maturation outweigh height gain, and may lead to permanent height deficit [9]. To improve the final adult height, several clinical strategies have been suggested, such as a gradual correction of hypothyroidism and/or adjunctive growth-promoting; the efficacy of these approaches is unconfirmed [9].

## 4. Conclusions

Autoimmune hypothyroidism is the most common thyroid disorder in children, usually presenting with typical symptoms, reported by the parents or the patient himself, which lead to an early diagnosis and replacement therapy. However, the classic symptoms may sometimes be not so clinically evident or recognizable, and this condition can be overlooked for years, which can have serious effects on pituitary secretion, multiple organs function and growth. Most of these manifestations are potentially reversible with levothyroxine therapy, but growth can be sometimes definitely compromised, especially in severe chronic forms.

This report describes the case of an adolescent with growth stunting which, by itself, is a classical presentation symptom of hypothyroidism. Instead, only the onset of myxedema, a very rare complication, led to the diagnosis of a long-standing hypothyroidism with impaired function of multiple organs and secondary pituitary hyperplasia. The diagnostic delay can be explained by the mild symptoms and the absence of extensive growth and puberty monitoring, in a context of a family history of delayed puberty. This case underlines the importance of pediatric check-ups for this age group; check-ups are essential for identifying conditions which, if not recognized and adequately treated, can have short- and long-term sequelae.

## Figures and Tables

**Figure 1 children-10-00614-f001:**
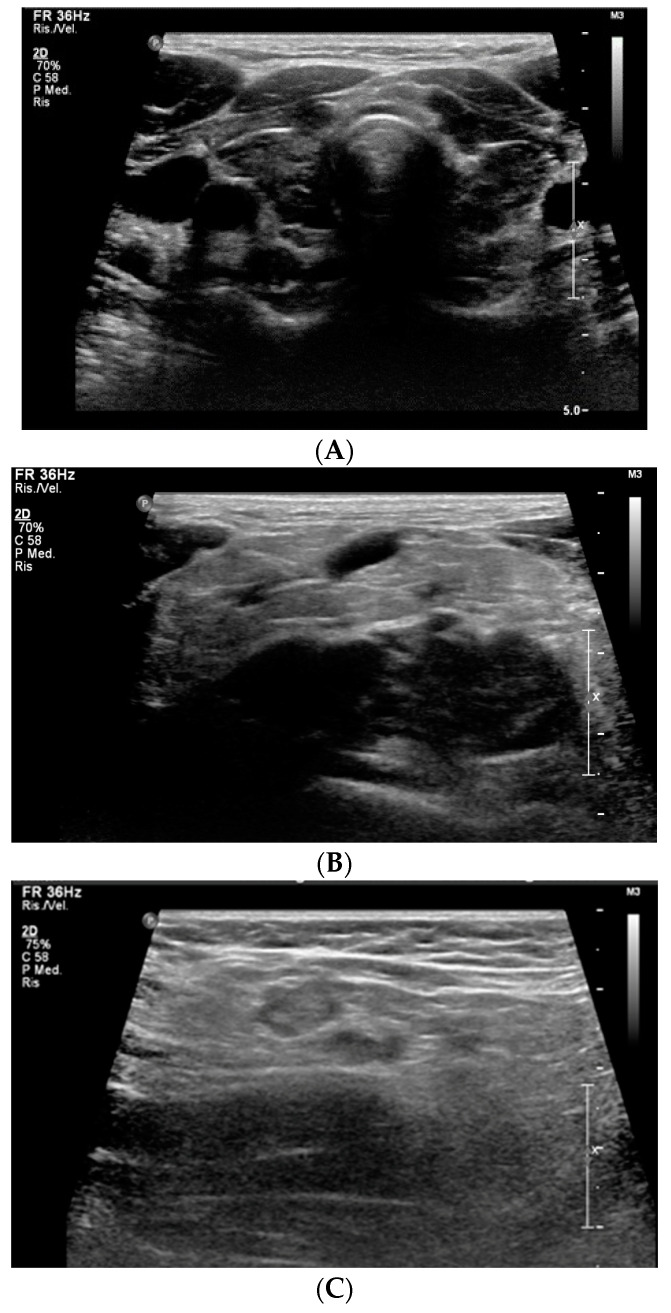
Ultrasound findings of thyroiditis (**A**), thickening and hyperechogenic of subcutaneous adipose tissue in the right supraclavicular region (**B**) and in the right armpit (**C**).

**Figure 2 children-10-00614-f002:**
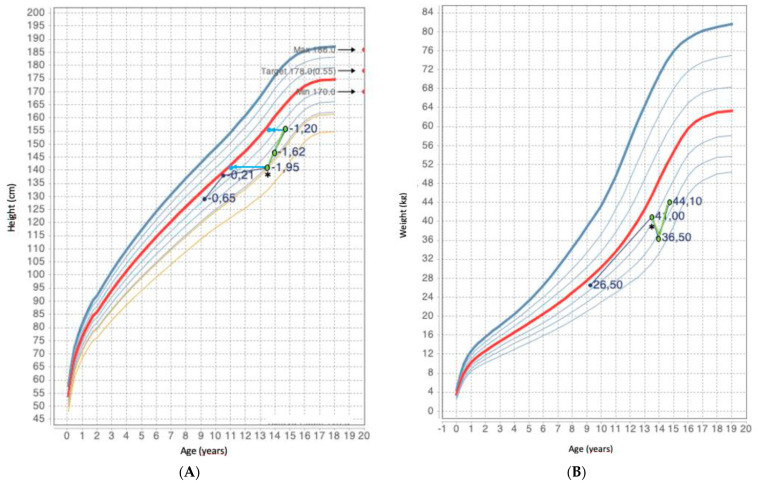
(**A**,**B**). Height and weight of the reported patient. Green spots represent height and weight at diagnosis, and at follow up in the endocrinology department. Dark blue spots show previous height and weight measurements based on pediatrician growth monitoring. Light blue spots represent the height projected based on the bone age, while light blue arrows reflect the difference with chronological age. Black asterisk indicates initiation of levothyroxine treatment at diagnosis. The red spots and marked interval correspond to the mid target height +/− 2 SDS.

**Figure 3 children-10-00614-f003:**
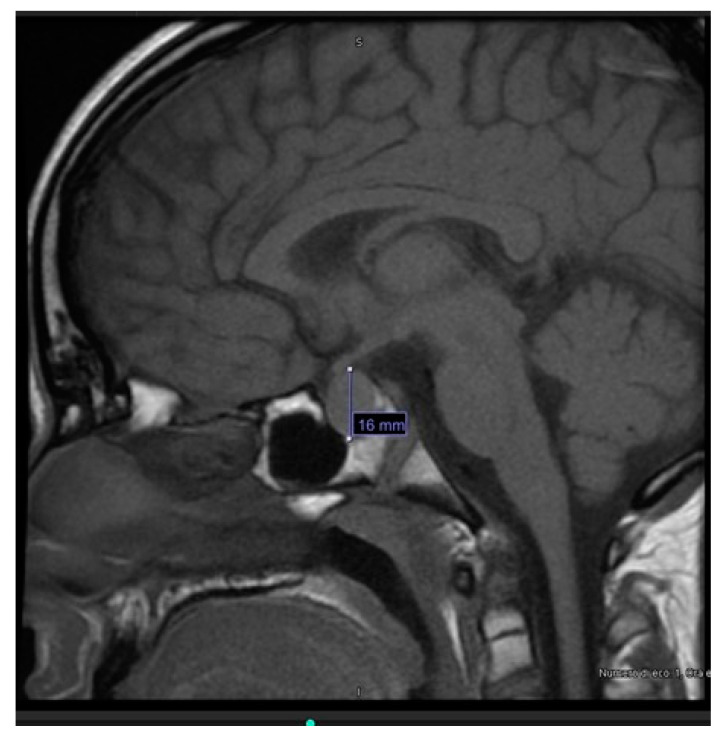
MRI of pituitary hyperplasia.

**Table 1 children-10-00614-t001:** Clinical and biochemical–hormonal data from diagnosis during follow-up.

	Diagnosis	at 2 Weeks	at 3 Weeks	at 2 Months	at 6 Months	at 1 Year
Clinical signs	Myxedema, dry and cold skin	Edema regression	Further improvement of clinical status			
Free T4 (pg/mL) (n.v. 9.8–16.3)	0.6	6.3	11.1	12.2	8	10
TSH μIU/mL)(n.v. 0.51–4.3)	>1000	610.9	41.9	1.02	1.04	0.72
Height (cm) SDS	141 −1.95				146.8 −1.62	155.6 −1.2
Bone age	11 years					13.5 years
Creatinine (mg/dL) (n.v. 0.47–0.73)	1.07	0.99	0.88	0.61	0.59	0.66
Prolactin (ng/mL) (n.v. 4–15.2)	118.4			36.2		15.1
Echocardiogram	Pericardial effusion up to 12 mm, not affecting cardiac hemodynamics	Pericardial effusion of 4 mm, not affecting cardiac hemodynamics			Regression of pericardial effusion	
MRI	Pituitary hyperplasia (16 × 12 × 16 mm)				Regression of pituitary hyperplasia	

## Data Availability

The research data for the current study are available from the corresponding author on reasonable request.

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
