# Peer review of "Pediatric Myxedema Due to Autoimmune Hypothyroidism: A Rare Complication of a Common Disorder"

_children, 2023, doi:10.3390/children10040614_

Round 1

Reviewer 1 Report (Previous Reviewer 2)

None, well revised

Author Response

Thank you. 

Reviewer 2 Report (Previous Reviewer 3)

Summary

This manuscript describes a boy with myxedema due to thyroiditis, a rare condition in children.  He had enough symptoms and signs to make the diagnosis almost certain and the rapid resolution with thyroxine treatment clinched it. 

Strengths

The description of the case is thorough, touching almost all facts relevant to hypothyroidism due to thyroiditis.  The manuscript is significantly improved from the previous version.

Areas for Improvement

The case description needs to be revised to fit the classical presentation sequence: history, physical exam, diagnostic evaluation, treatment and course.  Examples:

            lines 63-66 are history, not physical exam

            line 72: any “clinical” evaluation of the cardiac hemodynamics, needs to be part of the physical exam, not the diagnostic testing

            lines 95-96: Growth assessment is part of physical exam and currently appears after diagnostic testing.

            line 97: family history of delayed puberty is part of history and currently appears after diagnostic testing.

The manuscript has minor flaws in English usage and needs to be edited by someone whose primary language is English.  Examples:

            line 17: “the condition resulted consistent with myxedema”

            line 162: “in an age range (10-14 years) which is essential for monitoring growth and puberty”

            line 181: “in hypothyroid patients for transient renal impairment” should be “in hypothyroid patients with transient renal impairment” and similarly later in that paragraph.

            line 197: “growth stunting which, by itself, was to be a red flag for hypothyroidism.”

            line 201: “the lack of pediatric controls for growth and puberty monitoring”

As stated in the manuscript, “significant growth delay was present,” (line 19) which contradicts the start of that sentence, “The other symptoms of hypothyroidism were mild”.

In line 70, the statement “The electrocardiogram was normal” is so surprising with the pericardial infusion that you should consider adding, “including the voltages in all leads being normal for age” if that is a correct statement.

Minor points

The term for the medicine used to treat should be consistently either “levothyroxine” or “L-thyroxine”.  The journal editor may have a preference.

In the phrase, “clinical-hematochemical-instrumental improvement” I do not understand what “instrumental” means.

line 27: “such myxedema” should be “such as myxedema”

Fig 2B.  The black asterisk in Fig 2A should be added to Fig 2B.

line 106: “Light blue spots represent bone age at X-rays” should be “Light blue spots represent the height projected back to the bone age.”

line 151: “reported from the age of about 10.5 years” should be “demonstrated from the age of about 10.5 years.”  Had it been detected and “reported”, the diagnosis would probably have been made much sooner.

line 164: “hypocorticism has also to be ruled out” should be “hypocorticism has to be ruled out”.

I think the preferred term for “hypocorticism” is “hypocortisolism”.

line 174: “regresses with levothyroxine treatment, usually does not require surgery” should be “regresses with levothyroxine treatment and usually does not require surgery”

line 183: “pericardium ability” should be “pericardium’s ability”.

Author Response

Below the responses point-by-point:

1) The case description needs to be revised to fit the classical presentation sequence: history, physical exam, diagnostic evaluation, treatment and course.  Examples:

    • lines 63-66 are history, not physical exam
    • line 72: any “clinical” evaluation of the cardiac hemodynamics, needs to be part of the physical exam, not the diagnostic testing
    • lines 95-96: Growth assessment is part of physical exam and currently appears after diagnostic testing.
    • line 97: family history of delayed puberty is part of history and currently appears after diagnostic testing.

Response 1:

The case description has been revised respecting the classical order of the diagnostic process in the Emergency Room: family and patient history, physical exam (including clinical evaluation of the cardiac hemodynamics) and diagnostic evaluation. Only growth assessment was not included in the first physical exam, since the boy’s height was not measured in first instance in the emergency room: a complete auxological and pubertal assessment was performed after diagnosis, when the patient was admitted to the Endocrinology Department.

2) The manuscript has minor flaws in English usage and needs to be edited by someone whose primary language is English.  Examples:

    • line 17: “the condition resulted consistent with myxedema”
    • line 162: “in an age range (10-14 years) which is essential for monitoring growth and puberty”
    • line 181: “in hypothyroid patients for transient renal impairment” should be “in hypothyroid patients with transient renal impairment” and similarly later in that paragraph.
    • line 197: “growth stunting which, by itself, was to be a red flag for hypothyroidism.”
    • line 201: “the lack of pediatric controls for growth and puberty monitoring”

Response 2: English language has been revised throughout the manuscript and these sentences have been modified.

3) As stated in the manuscript, “significant growth delay was present,” (line 19) which contradicts the start of that sentence, “The other symptoms of hypothyroidism were mild”.

Response 3: The sentence has been modified.

4) In line 70, the statement “The electrocardiogram was normal” is so surprising with the pericardial infusion that you should consider adding, “including the voltages in all leads being normal for age” if that is a correct statement.

Response 4: The statement is correct and it has been added.

5) Minor points:

  • The term for the medicine used to treat should be consistently either “levothyroxine” or “L-thyroxine”.  The journal editor may have a preference.
  • In the phrase, “clinical-hematochemical-instrumental improvement” I do not understand what “instrumental” means.
  • line 27: “such myxedema” should be “such as myxedema”
  • Fig 2B.  The black asterisk in Fig 2A should be added to Fig 2B.
  • line 106: “Light blue spots represent bone age at X-rays” should be “Light blue spots represent the height projected back to the bone age.”
  • line 151: “reported from the age of about 10.5 years” should be “demonstrated from the age of about 10.5 years.”  Had it been detected and “reported”, the diagnosis would probably have been made much sooner.
  • line 164: “hypocorticism has also to be ruled out” should be “hypocorticism has to be ruled out”. I think the preferred term for “hypocorticism” is “hypocortisolism”.
  • line 174: “regresses with levothyroxine treatment, usually does not require surgery” should be “regresses with levothyroxine treatment and usually does not require surgery”
  • line 183: “pericardium ability” should be “pericardium’s ability”.

Response 5:

  • The term “levothyroxine” has now been employed in all the manuscript.
  • The phrase “clinical-hematochemical-instrumental improvement” has been replaced with “clinical, hematochemical and radiological improvement”, referring with term “radiological” to MRI and echocardiogram improvements
  • The listed sentences have been modified as suggested, the black asterisk has been added to Fig 2B and the term “hypocortisolism” has been chosen.

Reviewer 3 Report (New Reviewer)

It is a good case highlighting the importance of monitoring growth and puberty in children and the rare manifestations of severe untreated hypothyroidism. However, significant editing would need to be made for the language and grammar. 

Author Response

An extensive English revision was made by a native English-speaking colleague.

This manuscript is a resubmission of an earlier submission. The following is a list of the peer review reports and author responses from that submission.

Round 1

Reviewer 1 Report

The authors described a case of severe autoimmune hypothyroidism. There are a couple of concerns for this reviewer.

Major comments:

1) What is the new finding for the readers to know? Being rare differs from being novel.

2) The growth chart is not appropriately shown. Both height and weight are needed, and more plots should be illustrated.

Minor comments

1) Brain MRI is not necessary for this kind of case.

2) ECHO images of some other parts than the thyroid could be informative (new) for the readers.

3) There are several unnatural English. I will explain just the title.

    ・ It is not easy to understand what is rare. Myxedema in children is rare,

         or its development in autoimmune hypothyroidism is rare?

 ・’a severe autoimmune hypothyroidism' should be just 'severe

         autoimmune hypothyroidism.'

Reviewer 2 Report

This is a case report about a patient with severe untreated hypothyroidism secondary to Hashimoto’s thyroiditis, presenting as myxedema. This is not an original case but not commonly seen any more and therefore may be revisited.

1.      The English language has many grammar and typographic errors, should be extensively revised.

2.      Figure 1 and Table 1 are unnecessary and contribute very little to the manuscript and should be deleted.

3.      Abstract should be revised and shortened.

4.      ‘’Hyperprolactinemia, a very common laboratory finding of primary hypothyroidism, is due to increased TRH as a result of low thyroid hormone levels (T3 and T4). TRH stimulates prolactin as well as TSH secretion from pituitary gland and may cause galactorrhea in children with severe hypothyroidism. Prolactin level normalizes after thyroid hormone replacement’’ It might be helpful to add this point in the discussion section since high prolactin levels in the case presented, has been brought up several times throughout the manuscript.

5.      One other point is that, before treating severe hypothyroidism, laboratory work up should be done to exclude primary adrenal insufficiency since L-thyroxine replacement may precipitate adrenal crises in untreated Addison’s disease. This point should be added to the discussion section.

Reviewer 3 Report

Summary

This manuscript describes a boy with myxedema due to thyroiditis, a rare condition in children.  He had enough symptoms and signs to make the diagnosis almost certain and the rapid resolution of most signs with thyroxine treatment clinched it. 

Strengths

The description of the case is thorough, touching almost all facts relevant to hypothyroidism due to thyroiditis.

Areas for Improvement

If the manuscript is to be published in an English language journal, it needs to be edited by a native English speaking pediatric endocrinologist.

Table 1 should be overhauled.  The row titles in the left column should include Clinical presentation, free T4, TSH (NOT as a combined row), height, bone age, creatinine, etc.  Each should include the normal range or put the normal range as a second column.  The data under each time column does not need to repeat the title of the row.  Commentary e.g. “hyperprolactinemia” and “significant reduction of prolactin” do not belong in the table.  The last column is superfluous.

I do not think the growth chart enhances the manuscript.  The information is stated in the text and there are not enough data to warrant graphic presentation.

The presentation is not in the usual order.  Family history belongs before physical examination, not after the description of treatment.  The physical exam should start with height, weight, BMI and their percentiles, followed by pulse and blood pressure.  All of the symptoms should be in the Case Report.  Deep tendon reflexes, an abnormality of which is a hallmark of severe hypothyroidism, should be mentioned if they were performed.  “In our case, only mild constipation and slow speech were reported” is not mentioned until the Discussion.  The same applies to some of the negative history in the Discussion.

Minor points

Consider mentioning “pituitary crossover syndrome” (Google “precocious puberty and hypothyroidism” to see a bunch of references).  It likely explains why your patient had normal puberty despite his severe hypothyroidism.

I do not think “crepitus” is relevant to a skin examination.

It is hard to imagine that the electrocardiogram was normal based on the rest of the findings.  Were amplitudes compared to the normal values for age?  Likewise it is hard to imagine that the heart size on chest x-ray was normal despite a significant pericardial effusion.  12 mm on each side widens the heart shadow by 24 mm.  Were the cardiac hemodynamics assessed by doppler or clinically?

I do not understand what a “bulky” abdomen is.

“The other more common autoimmune diseases” is a mis-statement.  Most of the diseases listed are uncommon to rare in children.  Just remove “more common”.

Typo: “high of about 3.5 mm” should be “height of about 3.5 mm”

“have been programmed” is not part of usual English idiom.

I think “nowadays” is too folksy for a scientific publication.

I think the reason that “pituitary hyperplasia secondary to primary hypothyroidism” is uncommon is that there is probably no reason to perform an MRI in someone with very low free T4, very high TSH and normal visual fields.